

# Patent lifetime prediction using LightGBM with a customized loss

Jieming Liu[1], Peizhao Li[2] and Xiaowei Liu[3]

[1] School of Law, Humanities and Sociology, Wuhan University of Technology, Wuhan, Hubei, China
[2] School of Management, Wuhan University of Technology, Wuhan, Hubei, China
[3] Hubei Intellectual Property Research Center, Wuhan University of Technology, Wuhan, Hubei, China

## ABSTRACT

Patent lifespan is commonly used as a quantitative measure in patent assessments. Patent holders maintain exclusive rights by paying significant maintenance fees, suggesting a strong correlation between a patent's lifespan and its business potential or economic value. Therefore, accurately forecasting the duration of a patent is of great significance. This study introduces a highly effective method that combines LightGBM, a sophisticated machine learning algorithm, with a customized loss function derived from Focal Loss. The purpose of this approach is to accurately predict the probability of a patent remaining valid until its maximum expiration date. This research differs from previous studies that have examined the various stages and phases of patents. Instead, it assesses the commercial viability of individual patents by considering their lifespan. The evaluation process utilizes a dataset consisting of 200,000 patents. The experimental results show a significant improvement in the performance of the model by combining Focal Loss with LightGBM. By incorporating Focal Loss into LightGBM, its ability to give priority to difficult instances during training is enhanced, resulting in an overall improvement in performance. This targeted approach enhances the model's ability to distinguish between different samples and its ability to recover from challenges by giving priority to difficult samples. As a result, it improves the model's accuracy in making predictions and its ability to apply those predictions to new data.

## INTRODUCTION

When assessing patents, the length of time that a patent remains valid is used as a measurable indicator, often accompanied by information on forward citations (*Hikkerova, Kammoun & Lantz, 2014*). Considering that patent owners are required to regularly pay significant maintenance fees to maintain their exclusive rights, it is clear that the duration of a patent is directly linked to its commercial viability and economic worth. Empirical research consistently shows a robust positive correlation between the duration of a patent and its value. Therefore, the duration of a patent is often used as a measure of its quality in relation to its likelihood of achieving commercial success (*Guellec & van Pottelsberghe de la Potterie, 2000*). In essence, the longer a patent remains valid after it is first filed, the more likely it is to have direct or indirect economic importance (*Pakes, 1984*). Moreover, this

Corresponding author
Xiaowei Liu, wutiipteam@126.com

correlation is consistent with the behavior of patent holders, who tend to keep their patents when the advantages, whether direct or indirect, surpass the expenses associated with maintaining exclusive rights (*Hikkerova, Kammoun & Lantz, 2014*; *Serrano, 2010*). The complex relationship between the duration of a patent and its economic value highlights the crucial role that length of time plays in shaping the business environment of intellectual property.

Patent holders possess different strategic advantages in the market, utilizing their patents to accelerate the development of current technologies or products and surpass competitors (*Lemley & Shapiro, 2007*). By adopting this proactive approach, they are able to stay ahead of their competitors and strengthen their position in the market. In addition, patent holders have the ability to indirectly impact their competitors by impeding their business activities in the particular market that is protected by the patent (*Bader et al., 2012*). This establishes obstacles for new entrants and strengthens the market control of the patent holder. Moreover, forward citation data, which indicates the number of times a patent has been cited by subsequent patents (*Chen, 2017*), is a useful indicator of its inherent worth (*Narin, Noma & Perry, 1987*). These citations serve to confirm both the importance and originality of the patented technology, as well as its potential for future advancements and commercial success. Previous research has found a direct relationship between the value of a patent and the quantity of its forward citations. This information has been used to evaluate the value of a patent. In addition, several studies use the number of forward citations to determine the quality or technical influence of a patent (*Yoo, Lee & Won, 2006*). They also use this information to estimate the lifespan of the technology, predict future citations to assess the impact of the patent, analyze related technologies, and measure the technology cycle time index to understand technological depreciation.

Recently, aside from numerous applications of artificial intelligence and machine learning in financial (*Luo, Zhuo & Xu, 2023*; *Leow, Nguyen & Chua, 2021*) and business analytics (*Liu et al., 2024*; *Li & Sun, 2020*), there has been an increase in the use of machine learning techniques to assess the value or quality of patents. These approaches go beyond traditional methods of assessing patents based on their lifecycle and instead consider factors such as the monetary legal value of a patent, which can be determined by analyzing data from patent infringement lawsuits (*Lai & Che, 2009*). Additionally, these approaches also assess the transferability of patents, which is another aspect of patent quality (*Trappey et al., 2012*). Although previous studies have made significant progress in patent and technology evaluation, there are still areas that need further improvement (*Dai et al., 2024*; *Xu et al., 2024*). Prioritizing the evaluation of patent quality across various technologies has overshadowed the assessment of individual patents' quality or commercial potential in terms of their lifespan. However, it is important to acknowledge the clear connection between lifespan and commercial potential. Furthermore, previous research has relied heavily on forward citation data, which presents a practical challenge when analyzing newly submitted patents with limited or no forward citation information (*Yoo, Lee & Won, 2006*). Machine learning models developed on small datasets within specific technological fields may lack reproducibility when applied to patents in other fields. Therefore, further exploration and refinement are necessary to ensure broader applicability

(*Trappey et al., 2012*; *Lai & Che, 2009*). Although these models may perform exceptionally well in their specific area, applying them to different domains may necessitate further testing and adjustment.

In order to effectively address the specified limitations, this study aims to improve the comprehension of the viability of individual patents by evaluating their economic feasibility and forecasting the likelihood of their continued existence until their eventual expiration date. This research aims to transform the evaluation process by using an advanced machine learning model. It integrates different patent indicators that can be identified soon after filing to prioritize patents with the highest probability of long-term significance and value in the market. The rationale behind this approach is that these patents are more likely to generate profits, either directly or indirectly, throughout their entire lifespan. These indicators, as identified in relevant literature, demonstrate strong or possible connections with the duration of a patent's validity and, consequently, the commercial viability of a patent.

## BACKGROUND AND RELATED STUDIES

### USPTO patent lifetime

Ensuring the preservation of patent rights, as specified by the United States Patent and Trademark Office (USPTO), is an essential element of owning a patent that necessitates prompt attention and compliance with specific guidelines. After being granted a patent, the patent holder must promptly pay a maintenance fee in order to maintain their exclusive rights to the patent. The maintenance fees are required to be paid every 3–3.5 years, 7–7.5 years, and 11–11.5 years from the initial payment, as specified by the USPTO. It is crucial to emphasize that each fee has a grace period of six months. During this time, the payment must be made to prevent the patent from expiring and the subsequent loss of exclusive rights.

The magnitude of each maintenance fee may fluctuate depending on various factors, such as the number of claims, the nature of the applicant, and the specific payment term. Significantly, the fee for the third maintenance period, which takes place 12 years after the patent is granted, is generally higher in comparison to the fees for the earlier periods. Nonpayment of the maintenance fees within the designated time periods can lead to extra fees and, ultimately, the patent's termination.

If a patent expires because of non-payment, it is possible to restore the patent rights by filing a petition to the USPTO. Nevertheless, this procedure necessitates compelling evidence to establish that the failure to pay was not intentional. It is of utmost importance for patent holders to be watchful and proactive in handling their patent portfolios, making sure to meet maintenance fee deadlines in order to effectively protect their intellectual property rights.

Having a comprehensive understanding of patent term adjustments and expiration dates is crucial for both patent holders and applicants in the field of patent law. If a patent application encounters delays during the examination process, it may be granted a patent term adjustment. Despite this adjustment, the patent's validity and enforceability are

**Table 1   Code for USPTO maintenance fee event.**

| Event code | Description |
| --- | --- |
| EXP | The patent has lapsed due to non-payment of the maintenance fee |
| EXPX | Patent restored following payment of maintenance fee |
| M1551, M2551, M3551, M273, M283, M170, M173, M183 | Payment for the fourth year's maintenance fee |
| M1552, M2552, M3552, M274, M284, M171, M174, M184 | Payment for the eighth year's maintenance fee |
| M1553, M2553, M3553, M275, M285, M172, M175, M185 | Payment for the maintenance fee in the twelfth year |
| REM | A notice regarding the maintenance fee has been sent by mail |

preserved, but the maximum duration of the patent is limited to 17 years from the date it is issued. This adjustment is essential to ensure that patent holders are not unjustly disadvantaged as a result of delays caused by the examination process.

The duration of a patent's validity at USPTO can vary, with options ranging from 4, 8, 12, to 17 years from the issuance date, or 20 years from the filing date. The maximum duration of a patent is determined by either the 17-year period from the date of issuance or the 20-year period from the date of filing.

Examining the data on patent maintenance fee events provided by the USPTO provides valuable insights into the duration of patents (Table 1). Patents marked with a "EXP" event code have lapsed because the maintenance fees were not paid, indicating that their enforceable period has come to an end. In contrast, patents that have event codes indicating payment of the third-period fee indicate that they have been kept in force until their ultimate expiration date. Thus, patents that have received maintenance fee payments in the twelfth year after registration can be classified as patents that have either been sustained for their entire duration or are progressing towards that outcome.

Having a thorough grasp of patent term adjustments and expiration dates enables patent holders and stakeholders to effectively navigate the intricacies of patent maintenance. This ensures the protection of their intellectual property rights and allows them to maximize the value of their patents over their lifetime.

## Related studies

Various methodologies have been developed to evaluate the quality and impact of technologies or patents throughout their existence. *Bosworth & Jobome (2003)* measured the rate at which technology becomes outdated and charted the path of technical advancement. Their study utilized patent renewal data to monitor the longevity of patents and analyze statistical patterns concerning patent lifespan, with the goal of illustrating the entire lifespan of technologies from creation to becoming outdated. However, the act of exclusively assigning patents to a specific stage of the technology cycle may present difficulties, especially for emerging technologies that have limited information regarding the renewal of patents. Similarly, *Pakes & Schankerman (1984)* aimed to measure the rate at which

 

knowledge becomes outdated in the private sector by employing a model of patent renewal. They treated the patent value distribution as a probabilistic model. The application of this method, which relies on patent renewal data, may not be readily adaptable to rapidly evolving patented technologies in recent times. Our study differs by considering a wide range of indicators, such as applicant information, technical environment markers, and bibliographic data, in order to determine the value of a patent. This approach allows for a more thorough evaluation. The resilience of technologies was assessed by *Yoo, Lee & Won (2006)* through an analysis of forward citation data obtained from patents. Nevertheless, their research failed to consider various intrinsic and extrinsic bibliometric factors that may impact the duration of individual patents. Their methodology was based entirely on the premise that there is a strong correlation between the number of forward citations and the longevity of a technology. Therefore, this methodology is inadequate in fully capturing the various technological characteristics that are inherent to individual patents.

Many research projects have investigated the correlation between the duration of a patent and its perceived worth, frequently incorporating forward citation data. In their study, *Hikkerova, Kammoun & Lantz (2014)* analyzed 22,700 European patents and identified three distinct stages in the European patent lifecycle: procedural abandonment, natural abandonment, and late withdrawal. Their analysis emphasized the importance of a patent's age as a crucial factor in determining its increasing value over time. The researchers used multiple logistic regression models to analyze the relationship between different patent indicators and the probability of abandonment throughout the entire duration of the patent. *Fischer & Leidinger (2014)* proposed a link between auction prices and patent indicators. They used patent age, backward references, claims, and self-citations as variables in their study. Despite indicating a somewhat adverse effect of patent age on auction prices, the undeniable association between a patent's lifespan and its value remained. Nevertheless, there has been limited focus on examining the duration of recently registered or existing patents, which justifies the need for additional investigation into the projected lifespan of patented technologies using patent indicators.

Although many studies have examined the length of patents, only a few have specifically investigated the individual commercial value of patents by analyzing their lifespans. While there have been different approaches suggested by researchers to study the lifespan of patents or technology, many of these methods have limitations that restrict their effectiveness. These limitations include issues like lack of reproducibility or excessive dependence on lagging indicators such as forward citation data. Prior studies in this field have generally concentrated on a restricted dataset or specific technology sectors, hindering a comprehensive evaluation of patents across various technological domains. Furthermore, in order to compare outcomes across various technology sectors using methodologies specific to each technology, it is necessary to collect data repeatedly and conduct complex experiments. Unlike other methods, our approach includes patents from all technological fields and avoids using outdated measures, guaranteeing strong and universally applicable results.

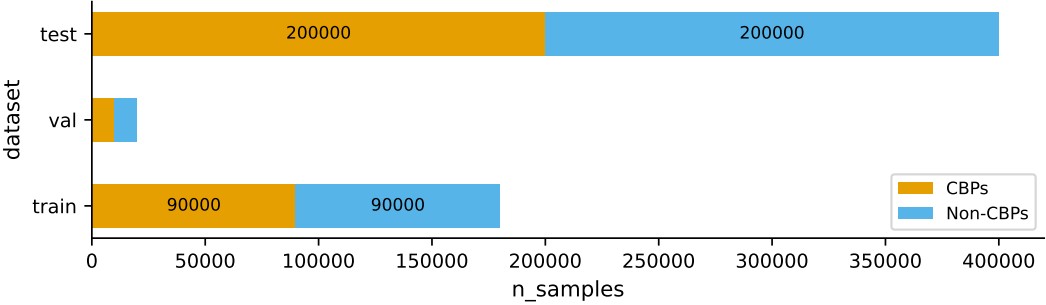

**Figure 1** **Distributions of the two classes CBPs and non-CBPs within each set of the data.**

# MATERIALS AND METHODS

## Data collection and processing

In this study, we used the dataset provided by *Choi et al. (2020)*. They designated a collection of patents that remained valid until their ultimate expiration dates as core business patents (CBPs). *Choi et al. (2020)* first created a database containing 2,715,519 patents that have been issued since 2000. The data was obtained from the USPTO bulk data, which was accessed through the website http://www.uspto.gov. This extensive database contains a wide range of patent-related information, including bibliographic data, citation information, and details of administrative processing. They accumulated a total of 1,277,662 patents that were issued after 2000 until February 2017. Patents issued before 2000 were not included in this count because there was a significant amount of missing information regarding their bibliographic and administrative data, making them different from the more recent patents. To guarantee the quality of the dataset, patents that had ambiguous bibliographic information were not included. As a result, a total of 952,408 patents with identifiable lifetimes were chosen. The ultimate dataset consisted of 278,512 patents reaching their fourth year of expiration, 233,869 patents reaching their eighth year of expiration, 126,869 patents reaching their twelfth year of expiration, and 313,419 patents retained for their maximum duration. Afterwards, a total of 200,000 patents were selected at random from the database to be used for training. The selection ensured an equal number of core business patents (CBPs) and non-CBPs, maintaining a ratio of 1:1. Non-CBPs were chosen regardless of their lifespans in order to reduce variations in the patent count across different time periods. *Choi et al. (2020)* further curated the remaining data by selecting 400,000 patents, maintaining a 1:1 ratio of CBPs to non-CBPs, to form a test set. In our research, we employed stratified sampling to partition their training set into our training set and validation set, with a distribution ratio of 90% and 10%, respectively. Additionally, we utilized their designated test set to evaluate the performance of our model. Figure 1 illustrates the distributions of the two classes within each set of the data used in our study.

## Patent indicators

The dataset consists of 24 indicators, as described in Table 2. Since all the data is in numerical format, there is no need for categorical to numerical conversion. Whether scaling is applied

**Table 2  Patent indicators provided in the data.**

| Indicator | Description |
| --- | --- |
| number_ipc | Number of Intellectual Property Claims (IPCs) contained within the patent |
| number_word_abstract | Total word count of the abstract in the patent |
| number_word_ft | Total word count of the patent's full text |
| number_citation | The count of cited patents by the patent as of the date of its issuance |
| number_citation_nation | Total count of different countries in the patent's backward citation data |
| number_priority | The quantity of priority patents that are linked to the given patent |
| number_priority_nation | Total number of different countries where the patent holds its priorities |
| number_claim_indep | Total count of independent claims within the patent |
| number_claim_dep | Total count of claims that rely on and further limit the scope of the main claim in the patent |
| number_claim_altered | Count of claims that have been either removed or included during the process of patent examination |
| number_applicant | Patent applicant count |
| number_foreign_applicant | Total count of non-domestic individuals or organizations of the patent |
| number_applicant_nation | Total count of different countries represented by patent applicants |
| number_assignee | Patent assignee count |
| number_avgword_indep | The average number of words in the patent's independent claims |
| average_gap_citation | The average duration between the issuance of a patent and its preceding patents |
| delivery_time | Duration of patent examination |
| number_family | The number of different countries in which the family patents of the patent are registered |
| number_foreign_family | Number of foreign family patents of the patent |
| number_family_nation | The quantity of distinct countries where the patent's family patents are located |
| ipc | 1 if a patent is classified under each of IPC sections A through H; 0 otherwise |
| ipc_activity | The annual average of the quantity of patents granted within the domain of technology |
| ipc_comp | The issuance year of patents granted by applicants in the technology field |
| ipc_size | The total number of patents granted in the field of technology over time |
| CBPs | CBP –the class label: 1 if the patent remains valid until its ultimate expiration date; 0 otherwise |

before feeding the data into a classifier depends on the underlying classification algorithm. We employed the light gradient-boosting machine (LightGBM) to construct our model for predicting whether a patent is a core business patent, given its reputation as one of the most effective classification algorithms for tabular data. For comparative analysis, we tested several other widely used algorithms, including the k-nearest neighbors (k-NN), support vector machines (SVM), random forests (RF), light gradient-boosting machine (LightGBM), eXtreme Gradient Boosting (XGBoost), and logistic regression (LR). It is important to emphasize that scaling is necessary for k-NN, SVM, and LR. There is no requirement to scale data for tree-based models such as LightGBM, RF, and XGBoost.

In addition, we also compare our model with TabTransformer (*Huang et al., 2020*), a sophisticated architecture for modeling tabular data in both supervised and semi-supervised learning environments. It employs self-attention based Transformers, with Transformer layers playing an important role in converting categorical feature embeddings into more robust contextual embeddings, resulting in improved prediction accuracy. The Tab Transformer model was configured with the following parameters: 32 channels, eight heads, six layers, and a dropout ratio of 0.32. A training process spanning 30 epochs was conducted, utilizing a batch size of 128 and a learning rate of 0.0001. The final model was chosen corresponding to the epoch at which the validation loss was minimum.

## LightGBM with focal loss

LightGBM stands out as a high-performance gradient boosting framework designed for efficiency and scalability in handling large-scale data. LightGBM, created by Microsoft, utilizes the innovative gradient-based one-side sampling and exclusive feature bundling techniques to improve training speed and minimize memory usage. The architecture of the system gives priority to the growth of the tree in a leaf-wise manner rather than in a level-wise manner. This enables faster convergence and reduces the amount of computational resources required. LightGBM is highly suitable for classification tasks involving tabular data due to its native support for categorical features and its effectiveness in handling imbalanced datasets. With widespread adoption in various domains, LightGBM has earned recognition for its impressive speed, accuracy, and versatility in model development and deployment scenarios.

LightGBM offers robust support for custom loss functions, enabling users to tailor the optimization process to specific needs or unique problem domains. This feature is especially valuable when conventional loss functions fail to sufficiently capture the intricacies of a specific task or when developers want to include domain-specific knowledge in the learning process. LightGBM enables users to specify their own loss functions, which facilitates the development of models that more closely correspond to the objectives and requirements of the problem. Whether it is minimizing classification errors, optimizing for precision and recall, or addressing other specific objectives, the flexibility of custom loss functions empowers users to fine-tune their models with precision and accuracy. LightGBM's ability to provide a versatile and customizable platform for machine learning tasks demonstrates its dedication to fostering innovation and enabling researchers and practitioners to effectively address complex challenges.

Our study employed Focal Loss (*Lin et al., 2020*) in conjunction with LightGBM to enhance the efficacy of our model. Assuming $p \in [0, 1]$ is the model's estimated probability for the class with label $y = 1$ ($y \in \{-1, 1\}$), the equation below provides the mathematical definition of the focal loss.

$$FL(p_t) = -a_t(1 - p_t)^\gamma \log(p_t), \tag{1}$$

where $p_t$ is a function depending on $y$ and $p$:

$$p_t = \begin{cases} p & \text{if } y = 1 \\ 1 - p & \text{otherwise,} \end{cases} \tag{2}$$

$\gamma$ is a modulating factor, and $p$ is computed by applying the sigmoid function to the raw margins $z$:

$$p = \frac{1}{1 + e^{-z}}. \tag{3}$$

$a_t$ is defined based on a weighting factor $\alpha$:

$$\alpha_t = \begin{cases} \alpha \in [0, 1] & \text{if } y = 1 \\ 1 - \alpha & \text{otherwise.} \end{cases} \tag{4}$$

The focal loss can be rewritten in terms of $y$ and $p$:

$$FL(y, p) = -\frac{y + 1}{2} \times \alpha(1 - p)^\gamma \log(p) - \frac{1 - y}{2} \times (1 - a)p^\gamma \log(1 - p). \tag{5}$$

In order to use a custom loss function with LightGBM, we need to compute its first and second order derivatives. The first order derivative of the focal loss is computed as:

$$\frac{\partial FL}{\partial z} = \alpha_t y (1 - p_t)^\gamma (p_t - 1 + \gamma p_t \log(p_t)). \tag{6}$$

Applying the chain rule, we have:

$$\frac{\partial FL}{\partial z} = \frac{\partial FL}{\partial p_t} \times \frac{\partial p_t}{\partial p} \times \frac{\partial p}{\partial z}. \tag{7}$$

After computing each part of the chain, we have:

$$\frac{\partial FL}{\partial z} = \alpha_t (1 - p_t)^\gamma \left( \frac{\gamma p_t \log(p_t) + p_{t-1}}{p_{t(1-p_t)}} \right) \times y \times p_{t(1-p_t)} \tag{8}$$
$$= \alpha_t y (\gamma p_t \log(p_t) + p_t - 1)(1 - p_t)^\gamma.$$

The second order derivative of the focal loss is computed as:

$$\frac{\partial^2 FL}{\partial z^2} = \frac{\partial}{\partial z} \left( \frac{\partial FL}{\partial z} \right) \tag{9}$$
$$= \frac{\partial}{\partial p_t} \left( \frac{\partial FL}{\partial z} \right) \times \frac{\partial p_t}{\partial p} \times \frac{\partial p}{\partial z}.$$

Using the following notations:

$$\frac{\partial FL}{\partial z} = u \times v, \tag{10}$$

$$u = \alpha_t y (1 - p_t)^\gamma, \tag{11}$$

$$v = \gamma p_t \log(p_t) + p_t - 1, \tag{12}$$

we have:

$$\frac{\partial u}{\partial p_t} = -\gamma \alpha_t y (1 - p_t)^{\gamma - 1}, \tag{13}$$

$$\frac{\partial v}{\partial p_t} = \gamma \log(p_t) + \gamma + 1. \tag{14}$$

Then, we can compute second order derivative as:

$$\frac{\partial^2 FL}{\partial z^2} = \left( \frac{\partial u}{\partial p_t} \times v + u \times \frac{\partial v}{\partial p_t} \right) \times \frac{\partial p_t}{\partial p} \times \frac{\partial p}{\partial z}. \tag{15}$$

The equations provided above will be utilized to calculate the first and second order derivatives necessary for computing the custom objective function applied in LightGBM.

## EXPERIMENTAL RESULTS AND DISCUSSION

We compared our model, LightGBM-FL, with TabTransformer, a state-of-the-art deep learning model for tabular data, and other machine learning models, including k-NN, SVM, LR, RF, XGBoost, and LightGBM with the default loss function. We chose the area under the receiver operating characteristic curve (AUC-ROC) as the primary evaluation metric. We also reported other common metrics, including accuracy, sensitivity (recall), specificity, precision, and F1-score.

Table 3 presents the performance of all models on the test set. The performance of our model, LightGBM-FL, is significantly better than other models across almost all evaluation metrics, particularly in the primary metric, AUC-ROC. The performance of the TabTransformer model was unsatisfactory due to its dependence on categorical data for utilizing contextual embedding and attention mechanisms. However, that data type was missing from the dataset, resulting in a decrease in its effectiveness.

Figure 2 shows the ROC curves, which were computed on the test set, for all of the models. The figure clearly demonstrates that LightGBM-FL outperforms all other models in terms of performance. Following closely are two gradient boosting algorithms: LightGBM with the default loss function and XGBoost. Their performances are strikingly comparable, as evidenced by their nearly identical ROC curves. Figure 3 displays a similar trend, showing the Precision-Recall curves calculated on the test set for all models.

Our experiments demonstrate that incorporating Focal Loss with LightGBM significantly improves the model's performance. The utilization of Focal Loss in LightGBM has improved the ability to prioritize difficult instances during training. Focal Loss is particularly effective in dealing with real-world datasets by assigning less weight to

**Table 3** Performance on the test set of all the models (bold indicates the highest value for each metric).

| Model | AUC-ROC | Accuracy | Sensitivity | Specificity | Precision | F1-score |
|---|---|---|---|---|---|---|
| k-NN | 0.7161 | 0.6641 | 0.6869 | 0.6413 | 0.6569 | 0.6716 |
| SVM | 0.7491 | 0.6818 | 0.7011 | 0.6626 | 0.6751 | 0.6878 |
| LR | 0.7509 | 0.6853 | 0.7172 | 0.6535 | 0.6742 | 0.6951 |
| RF | 0.7949 | 0.7229 | 0.7909 | **0.6548** | 0.6962 | 0.7405 |
| XGBoost | 0.8076 | 0.7347 | 0.8446 | 0.6248 | 0.6924 | 0.7610 |
| TabTransformer | 0.7241 | 0.6032 | 0.3801 | 0.8264 | 0.6864 | 0.4892 |
| LightGBM | 0.8082 | 0.7354 | 0.8456 | 0.6253 | 0.6929 | 0.7616 |
| LightGBM-FL | **0.8185** | **0.7447** | **0.8577** | 0.6318 | **0.6997** | **0.7706** |

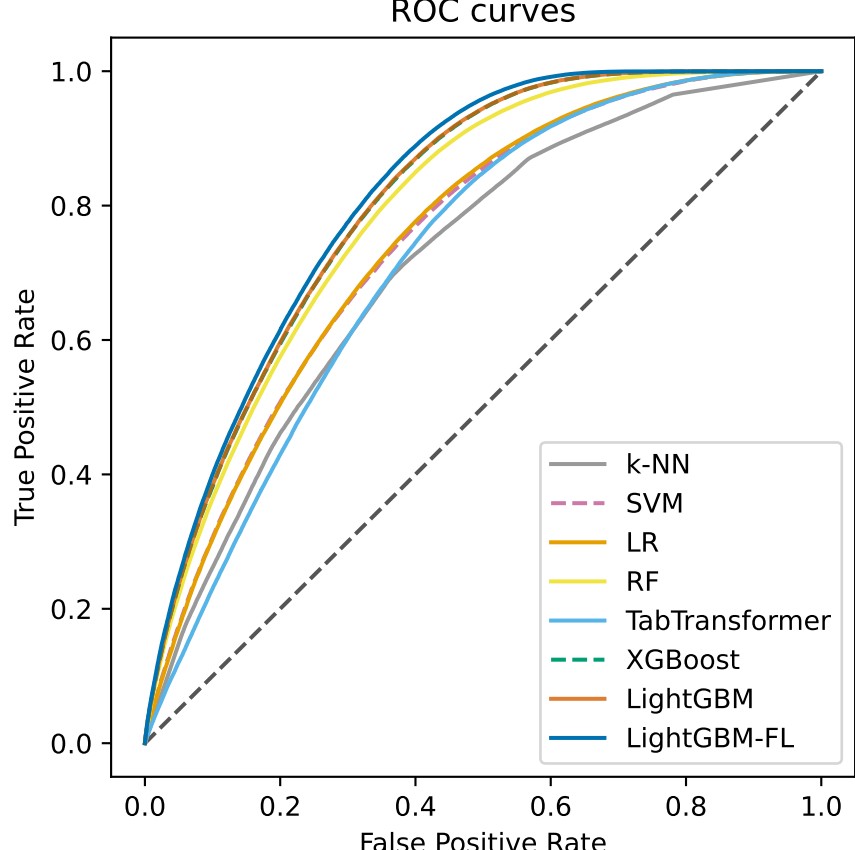

**Figure 2** The ROC curves computed on the test set of all the models.

well-classified examples. The implementation of this focused strategy has successfully enhanced the overall performance of the model. LightGBM with Focal Loss enhances its discriminative ability and robustness by prioritizing difficult-to-classify samples. This ultimately leads to superior predictive accuracy and generalization capabilities.

Figure 4 displays the top 15 features that have been identified by our model, LightGBM-FL. Among the features considered, `ipc_size`, `ipc_comp`, and `ipc_activity` stand out as

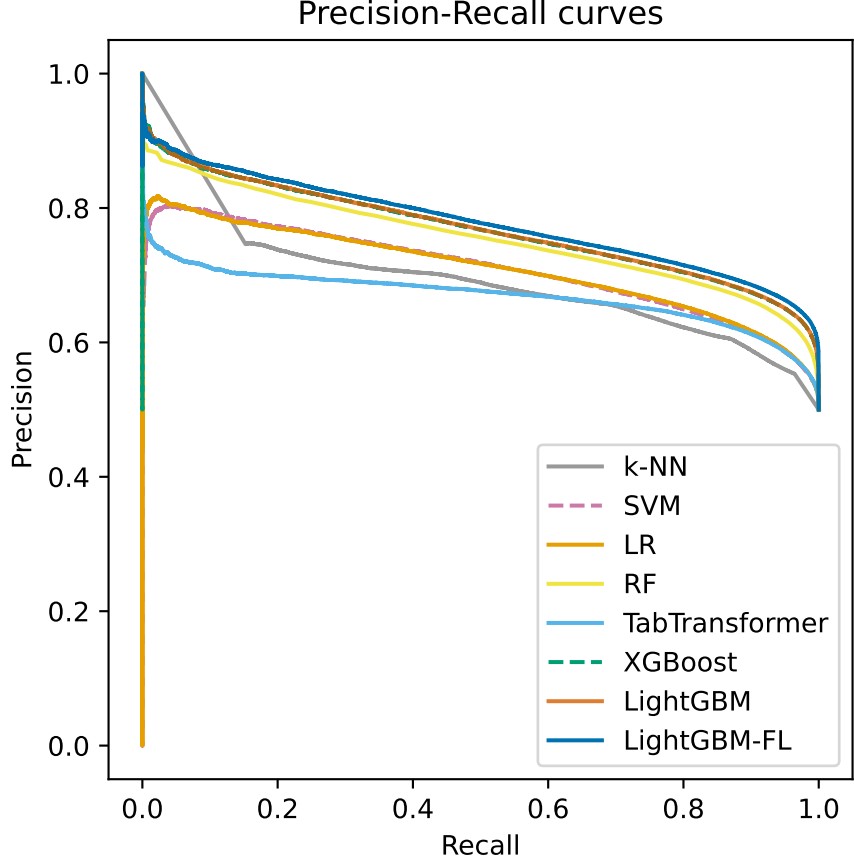

**Figure 3** **The precision-recall curves computed on the test set of all the models.**

the most significant in determining whether a patent will be maintained until its maximum expiration date, far exceeding the importance of other features.

The IPC size (`ipc_size`) is a measure that provides a comprehensive assessment of the extent and range of technology fields. It is calculated by taking the average number of patents issued within each field. This method was described by *Lai & Che (2009)* and by *Guellec & van Pottelsberghe de la Potterie (2000)*. The IPC competitiveness(`ipc_comp`) measures the level of competition among patent holders in particular fields by counting the number of applicants with patents in those fields (*Fabry et al., 2006*; *Guellec & van Pottelsberghe de la Potterie, 2000*). In addition to these metrics, the IPC activity(`ipc_activity`) measures the rate at which patents are issued each year in different technology fields. This provides insight into the level of innovation and vitality within those domains (*Ernst, 2003*). These environmental indicators have a significant impact on the duration of patents and influence the overall landscape of patent duration.

The findings of our study emphasize the importance of these three metrics, corroborating the results of *Choi et al. (2020)*, where they were also identified as part of the five selected features. The alignment emphasizes the significant importance of IPC size, IPC competitiveness, and IPC activity in comprehending the durability and fluctuations of

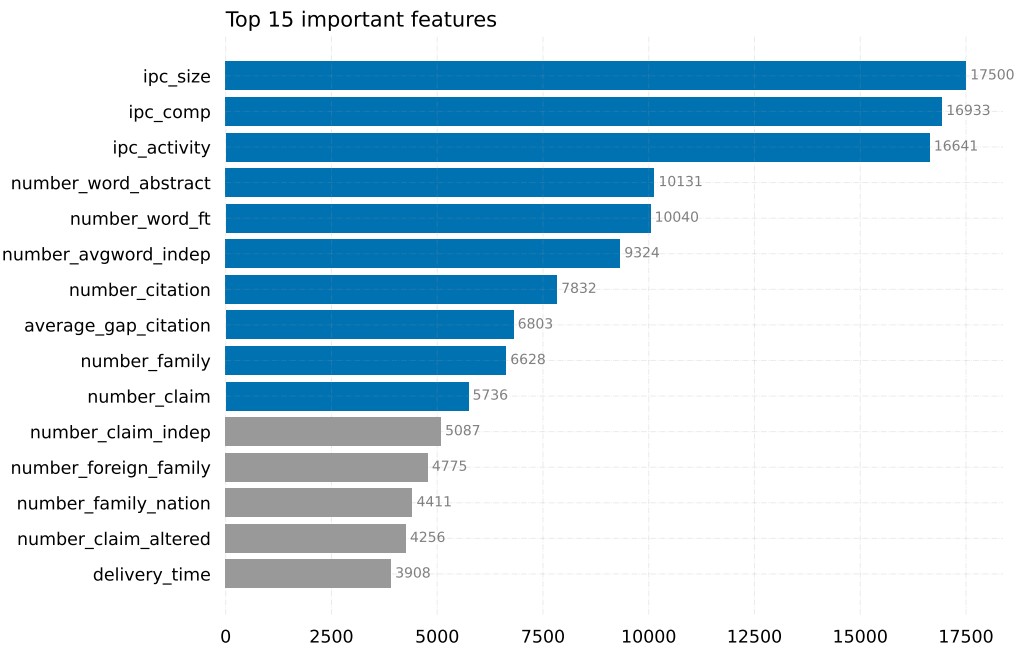

**Figure 4** Top features in our LightGBM-FL model (the top 10 are in blue).

patents across different technology domains. By integrating these metrics into our analysis, we acquire valuable insights into the interaction between environmental factors and patent duration, enhancing our comprehension of the intricate connection between technology fields and the longevity of patents.

## CONCLUSIONS

This study presents a novel method that combines LightGBM, a sophisticated algorithm, with a customized loss function inspired by Focal Loss. The objective is to precisely forecast the likelihood of a patent enduring until its ultimate expiration date. It is crucial to emphasize patents that lead to prolonged survival because they are anticipated to yield steady profits for the entire duration of their validity. Our research differs from previous studies that focused on the lifespan and stages of patents or technologies. Instead, we assess the individual business potential of patents by taking into account their longevity. Our methodology offers a distinct approach to evaluating the commercial viability of a patent by forecasting its likelihood of maintaining validity until its ultimate expiration date. The results of our experiment clearly show a substantial improvement in the performance of the model by combining Focal Loss with LightGBM. By utilizing Focal Loss in LightGBM, the model's capacity to give priority to difficult instances during training is enhanced, particularly when dealing with real-world datasets where accurately classified examples are given lower weights. The concentrated strategy improves the overall effectiveness of the model, enhancing its ability to distinguish and withstand challenging samples by giving

them higher priority. LightGBM with Focal Loss demonstrates superior predictive accuracy and generalization capabilities.

### Funding
The authors received no funding for this work.

### Competing Interests
The authors declare there are no competing interests.

### Author Contributions
- Jieming Liu conceived and designed the experiments, performed the experiments, analyzed the data, performed the computation work, prepared figures and/or tables, authored or reviewed drafts of the article, and approved the final draft.
- Peizhao Li conceived and designed the experiments, performed the experiments, analyzed the data, performed the computation work, prepared figures and/or tables, authored or reviewed drafts of the article, and approved the final draft.
- Xiaowei Liu conceived and designed the experiments, performed the experiments, analyzed the data, performed the computation work, prepared figures and/or tables, authored or reviewed drafts of the article, and approved the final draft.

### Data Availability
The Python code and data used in the study are available in the Supplemental File.

### Supplemental Information
Supplemental information for this article can be found online at http://dx.doi.org/10.7717/peerj-cs.2044#supplemental-information.

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
