# Peer review of "Patent lifetime prediction using LightGBM with a customized loss"

_PeerJ Computer Science, doi:10.7717/peerj-cs.2044_

## Round 0.1 · original submission · Major Revisions

Please revise the manuscript to address the comments from the reviewers, especially considering implementing more machine learning and deep learning algorithms for comparison.

**Language Note:** The review process has identified that the English language must be improved. PeerJ can provide language editing services - please contact us at [email protected] for pricing (be sure to provide your manuscript number and title). Alternatively, you should make your own arrangements to improve the language quality and provide details in your response letter. – PeerJ Staff

Reviewer 1 ·

Basic reporting

The manuscript was well-written, using clear and unambiguous technical language. The introduction and background sections provide adequate context for the study, and all relevant prior literature is properly referenced throughout the manuscript. The figures are of high quality, vector format, and well-presented in the manuscript. They are relevant to the content of the manuscript and have been appropriately described and labeled.

The code and data were distributed as a supplemental document, making it easily accessible and reproducible. The manuscript is "self-contained" and covers all important results, and it clearly matches the scope of the journal.

Another positive aspect is that the authors give a thorough explanation of patent lifetime. They also include the formulas for the Focal Loss along with proofs, which makes it easier to understand how to implement it.

Experimental design

The research question is defined clearly, and the study makes a valuable contribution to the field. The authors are clever when using LightGBM, a top machine learning algorithm that supports customized losses, combined with an advanced loss function in computer vision. The implementation in the code matches the provided formulas.

However, the study could be improved further if more algorithms are tested and compared. For example, some advanced deep learning architectures, e.g., the Tab Transformer, have shown promising performances on tabular datasets.

Validity of the findings

An additional experiment, as proposed above, could be carried out to further demonstrate the technical soundness of the results. The conclusions were well-stated, related to the original question investigated, and limited to those supported by the experimental results.

Additional comments

A revision is needed for this manuscript.

Cite this review as

Reviewer 2 ·

Basic reporting

The manuscript looks well-structured and written in a formal English manner. The literature study is adequate and provides a thorough overview of the topic. The tables and figures are well-designed and represent the research findings in a clear manner. The quality of the manuscript could be improved by including additional visualizations and addressing specific points for clarification, as outlined below.

Experimental design

The manuscript describes the method in a clear and technically sound manner. The proposed approach is compelling, as it makes use of LightGBM, a top classification algorithm for tabular data, and builds on the success of Focal Loss in computer vision. The study should be reproducible, as the code and data were provided. Upon reviewing the code, it appears to be correct. However, to further improve the manuscript, authors are encouraged to include additional information on the following aspects:
- More machine learning algorithms, such as k-NN and SVM, should be evaluated for comparison purposes.
- Given that the authors have already provided the ROC curves, they may want to include the Precision-Recall curves for the tested models as well.

Validity of the findings

The findings appear to be valid and have potential value for academics from various disciplines, particularly those interested in forecasting patient lifespans and applying such insights to business analytics. By including both the dataset and the code in the study, replication and further research are made easier. It would be beneficial for the authors to delve deeper into related studies that use machine learning in similar domains.

Additional comments

Please comment on the statistical aspects and confirm whether a statistical analysis was performed or is required for the study.

Cite this review as

---

## Round 0.2 · accepted · Accept

We are pleased to inform you that your manuscript has been accepted for publication following a thorough review by two independent reviewers, both of whom recommended acceptance.

Reviewer 1 ·

Basic reporting

- No more comments.

Experimental design

- The authors conducted additional experiments to assess additional machine learning algorithms in conjunction with the Tab Transformer, a state-of-the-art deep learning architecture designed for tabular data. Surprisingly, the Tab Transformer did not perform well. However, the authors' explanation makes sense, as Tab Transformer relies on categorical data to use contextual embedding and attention mechanisms, which was missing from the dataset, reducing its effectiveness.
- The experiments were well-designed, and I have no more comments.

Validity of the findings

- No more comments.

Additional comments

- This revision can be accepted for publication.

Cite this review as

Reviewer 2 ·

Basic reporting

No comment

Experimental design

As per my suggestion, the authors have conducted additional experiments and included more visualizations in the revision. I have no further comments.

Validity of the findings

No comment

Additional comments

The manuscript can be accepted.

Cite this review as